# Association between exercise habits and stroke, heart failure, and mortality in Korean patients with incident atrial fibrillation: A nationwide population-based cohort study

Hyo-Jeong Ahn[1]◎, So-Ryoung Lee[1]◎, Eue-Keun Choi[1,2]*, Kyung-Do Han[3], Jin-Hyung Jung[4], Jae-Hyun Lim[1], Jun-Pil Yun[1], Soonil Kwon[1], Seil Oh[1,2], Gregory Y. H. Lip[2,5,6]

1 Department of Internal Medicine, Seoul National University Hospital, Seoul, Republic of Korea,
2 Department of Internal Medicine, Seoul National University College of Medicine, Seoul, Republic of Korea,
3 Department of Statistics and Actuarial Science, Soongsil University, Seoul, Republic of Korea,
4 Department of Medical Statistics, College of Medicine, Catholic University of Korea, Seoul, Republic of Korea, 5 Liverpool Centre for Cardiovascular Science, University of Liverpool and Liverpool Heart and Chest Hospital, Liverpool, United Kingdom, 6 Department of Clinical Medicine, Aalborg University, Aalborg, Denmark

◎ These authors contributed equally to this work.
* choiek17@snu.ac.kr

**Data Availability Statement:** All relevant data are within the manuscript and its Supporting Information files.

## Abstract

### Background

There is a paucity of information about cardiovascular outcomes related to exercise habit change after a new diagnosis of atrial fibrillation (AF). We investigated the association between exercise habits after a new AF diagnosis and ischemic stroke, heart failure (HF), and all-cause death.

### Methods and findings

This is a nationwide population-based cohort study using data from the Korea National Health Insurance Service. A retrospective analysis was performed for 66,692 patients with newly diagnosed AF between 2010 and 2016 who underwent 2 serial health examinations within 2 years before and after their AF diagnosis. Individuals were divided into 4 categories according to performance of regular exercise, which was investigated by a self-reported questionnaire in each health examination, before and after their AF diagnosis: persistent non-exercisers (30.5%), new exercisers (17.8%), exercise dropouts (17.4%), and exercise maintainers (34.2%). The primary outcomes were incidence of ischemic stroke, HF, and all-cause death. Differences in baseline characteristics among groups were balanced considering demographics, comorbidities, medications, lifestyle behaviors, and income status. The risks of the outcomes were computed by weighted Cox proportional hazards models with inverse probability of treatment weighting (IPTW) during a mean follow-up of 3.4 ± 2.0 years. The new exerciser and exercise maintainer groups were associated with a lower risk of HF compared to the persistent non-exerciser group: the hazard ratios (HRs) (95% CIs)

**Funding:** This work and EKC, as a representative of all authors (HJA, SRL, KDH, JHJ, JHL, JPY, SK, SO, and GYHL), was supported by the Korea Medical Device Development Fund grant funded by the Korea government (the Ministry of Science and ICT, the Ministry of Trade, Industry and Energy, the Ministry of Health & Welfare, Republic of Korea, the Ministry of Food and Drug Safety) (Project Number: HI20C1662), and by the Korea National Research Foundation funded by the Ministry of Education, Science and Technology (Grant 2020R1F1A106740). The funders had no role in study design, data collection and analysis, decision to publish, or preparation of the manuscript.

**Competing interests:** I have read the journal's policy and the authors of this manuscript have the following competing interests: HJA, SRL, KDH, JHJ, JHL, JPY, SK, SO: None to disclose. EKC: Research grants from Bayer, BMS/Pfzer, Biosense Webster, Chong Kun Dang, Daiichi-Sankyo, Samjinpharm, Sanof-Aventis, Seers Technology, Skylabs, and Yuhan. GYHL: Consultant for Bayer/Janssen, BMS/Pfzer, Medtronic, Boehringer Ingelheim, Novartis, Verseon and Daiichi-Sankyo. Speaker for Bayer, BMS/Pfzer, Medtronic, Boehringer Ingelheim, and Daiichi-Sankyo. No fees are received personally. The external funders and sponsors of the study had no role in study design and conduct of the study; in the collection, analysis, and interpretation of the data; in the preparation, review, or approval of the manuscript; or in the decision to submit the manuscript for publication.

**Abbreviations:** AF, atrial fibrillation; ASD, absolute standardized difference; CCI, Charlson Comorbidity Index; CKD, chronic kidney disease; COPD, chronic obstructive pulmonary disease; CVD, cardiovascular disease; HF, heart failure; HR, hazard ratio; ICD-10-CM, International Classification of Diseases–10th Revision, Clinical Modification (ICD-10-CM); IPAQ, International Physical Activity Questionnaire; IPTW, inverse probability of treatment weighting; MET, metabolic equivalent of task; MI, myocardial infarction; NHID, National Health Information Database; PAD, peripheral artery disease; PY, person-years.

were 0.95 (0.90–0.99) and 0.92 (0.88–0.96), respectively ($p < 0.001$). Also, performing exercise any time before or after AF diagnosis was associated with a lower risk of mortality compared to persistent non-exercising: the HR (95% CI) was 0.82 (0.73–0.91) for new exercisers, 0.83 (0.74–0.93) for exercise dropouts, and 0.61 (0.55–0.67) for exercise maintainers ($p < 0.001$). For ischemic stroke, the estimates of HRs were 10%–14% lower in patients of the exercise groups, yet differences were statistically insignificant ($p = 0.057$). Energy expenditure of 1,000–1,499 MET-min/wk (regular moderate exercise 170–240 min/wk) was consistently associated with a lower risk of each outcome based on a subgroup analysis of the new exerciser group. Study limitations include recall bias introduced due to the nature of the self-reported questionnaire and restricted external generalizability to other ethnic groups.

## Conclusions

Initiating or continuing regular exercise after AF diagnosis was associated with lower risks of HF and mortality. The promotion of exercise might reduce the future risk of adverse outcomes in patients with AF.

## Author summary

### Why was this study done?

- Atrial fibrillation (AF) is associated with an increased risk of stroke, heart failure, and death. As AF and its related healthcare burden are expected to surge, integrated management of AF is advocated as part of holistic care.

- Exercise has been established to benefit AF-related outcomes, including symptoms, incidence, recurrence, burden, and quality of life. However, there are no current data providing the association between exercise and cardiovascular morbidities in patients with AF.

### What did the researchers do and find?

- We performed a retrospective analysis of 66,692 patients with newly diagnosed AF who underwent 2 serial health examinations before and after their AF diagnosis from 2010 to 2016 using data from the Korea National Health Insurance Service.

- Individuals self-reported exercise status by a questionnaire included in each health examination and were categorized into 4 groups according to their change of exercise status from before to after their AF diagnosis. We investigated and compared the incidences of ischemic stroke, heart failure, and all-cause death across the groups.

- Initiating or continuing regular exercise after a diagnosis of AF was associated with a 5%–8% lower risk of HF and 17%–39% lower risk of mortality than being a persistent non-exerciser. For ischemic stroke, the estimated hazard ratios were 10%–14% lower in patients in the exercise group, yet statistical significance was undetermined.

**What do these findings mean?**

- We suggest the potential benefits of exercise as a lifestyle intervention for cardiovascular outcomes in AF patients by demonstrating the association. The promotion of exercise might reduce the future risk of adverse outcomes in patients with AF.

## Introduction

Atrial fibrillation (AF) is the most common sustained cardiac arrhythmia, and the prevalence has been continuously growing over the past decades, potentially due to aging and an increase of risk factors predisposing to AF [1–3]. The clinical implication of AF is that it is associated with an increased risk of stroke, heart failure (HF), myocardial infarction (MI), and death [2,4]. As AF and its related healthcare burden are expected to surge in the coming years [5], this common condition necessitates a more integrated approach to management.

Many studies have suggested risk factor modification be incorporated as the fourth pillar of AF management along with anticoagulation, rhythm control, and rate control [6–8]. Exercise of regular moderate intensity, as a part of risk factor management of AF, not only improves underlying conditions related to atherosclerotic cardiovascular disease (CVD) but has been established to provide benefit for AF-related outcomes, including symptoms, incidence, recurrence, burden of AF, and quality of life [9–12]. Though the link between exercise and AF-related outcomes has been supported by substantial evidence, benefits in cardiovascular morbidities led by exercise in AF patients are less likely to be offered. The majority of deaths in AF patients are cardiovascular in origin [13,14], thus emphasizing that optimal management of underlying heart disease should be a part of the holistic approach to AF care [15].

Given the paucity of information about cardiovascular outcomes related to exercise habit change after a new diagnosis of AF, we aimed to investigate the association between exercise habits after a new AF diagnosis and the risk of ischemic stroke, HF, and all-cause death.

## Methods

### Data source and study population

We defined a population-based cohort from the National Health Information Database (NHID), which incorporates all data from the National Health Insurance Service and covers the entire population of the Republic of Korea (hereafter denoted as Korea). NHID is only accessible online at designated analysis centers, with formal payment according to the period of browsing and analyzing the data and applying strict regulations regarding data release (https://nhiss.nhis.or.kr/). All insured adults are eligible for a biennially conducted general health examination, and examination reports—including demographic variables, self-questionnaire survey, clinical laboratory values, inpatient and outpatient usage, prescription records, income-based insurance contributions, and date of death—are available from NHID [16,17]. The institutional review board at Seoul National University Hospital (E-2001-111-1096) authorized this study. This study is reported as per the Strengthening the Reporting of Observational Studies in Epidemiology (STROBE) guideline (S1 STROBE Checklist). The statistical analysis plan was not prespecified. However, the analysis was performed with a priori–defined outcomes and covariates that were validated based on our previous publications before analysis [18,19].

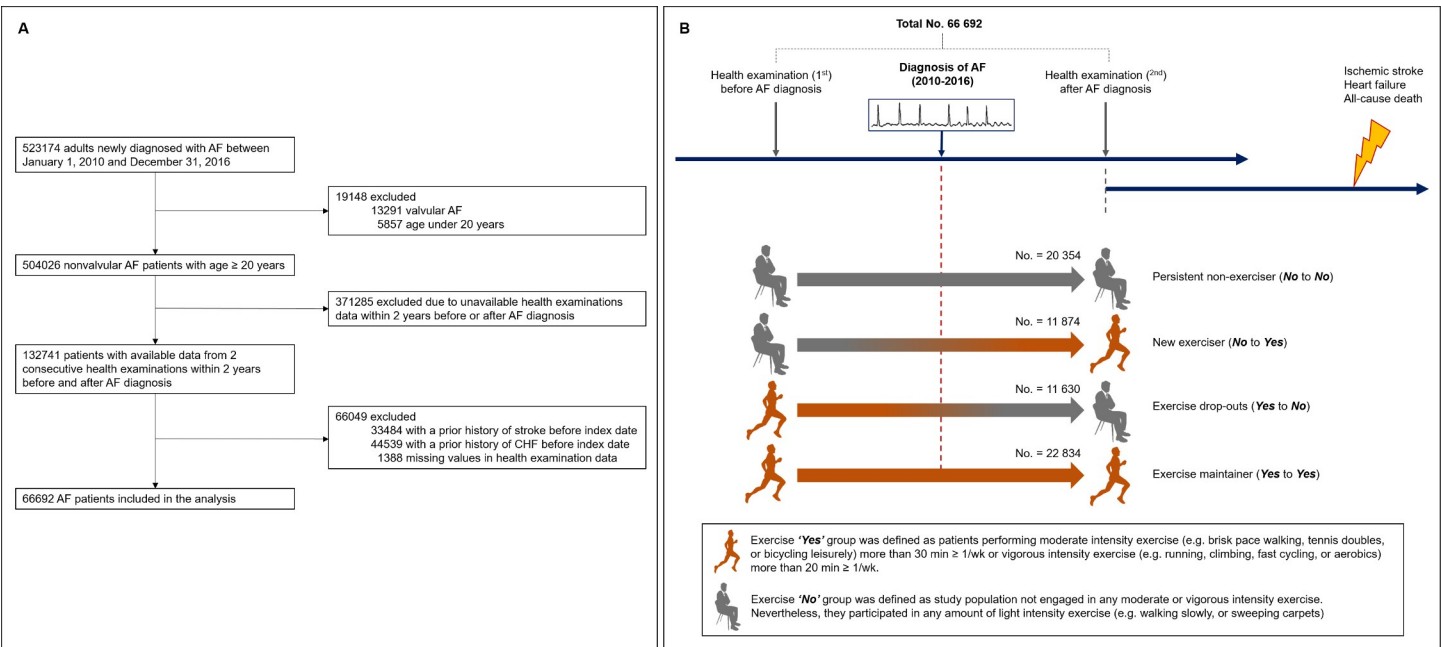

**Fig 1. Selection and categorization of the study population.** (A) Selection of study population from the National Health Insurance Service. (B) Categorization of study population according to the change of exercise status and overall scheme of study, including clinical outcomes. AF, atrial fibrillation; CHF, congestive heart failure.

We identified 523,174 patients newly diagnosed with AF between January 1, 2010, and December 31, 2016, and excluded patients with valvular AF or age under 20 years. Among the remaining patients, 132,741 patients who had data from 2 consecutive health examinations within 2 years before and after AF diagnosis were analyzed. Since we evaluated de novo incidence of ischemic stroke and HF after AF diagnosis, patients with prior recorded history of ischemic stroke or HF before the index date (health examination date after AF diagnosis) were excluded. Finally, 66,692 patients with AF were included in the analysis (Fig 1A).

## Exercise evaluation

The individuals self-reported intensity and frequency of exercise via a questionnaire about lifestyle behaviors at the serial health examinations before and after AF diagnosis. The survey structure used for exercise evaluation was adopted and modified from the International Physical Activity Questionnaire (IPAQ), which was developed by the World Health Organization (WHO); both the reliability and validity of this short-form survey were reported in previous studies [20–23]. The questionnaire section on exercise was composed of 3 questions asking about the frequency (days per week) of light, moderate, and vigorous exercise during a recent week, i.e., the frequency of (i) light intensity exercise (e.g., walking slowly or sweeping carpets) for more than 30 min, (ii) moderate intensity exercise (e.g., brisk-pace walking, tennis doubles, or bicycling leisurely) for more than 30 min, and (iii) vigorous intensity exercise (e.g., running, climbing, fast cycling, or aerobics) for more than 20 min. Exercise "Yes" was defined as performing moderate intensity exercise for more than 30 min or vigorous intensity exercise for more than 20 min, at least once a week. Exercise "No" was defined as not engaging in any moderate or vigorous intensity exercise, regardless of any amount of light intensity exercise. By comparing categorization of exercise status at the health examinations before and after AF diagnosis, the study population was divided into 4 groups: persistent non-exercisers (No to No), new exercisers (No to Yes), exercise dropouts (Yes to No), and exercise maintainers (Yes

to Yes). Fig 1B summarizes our study's overall configuration and the selection of the population included in the analysis.

To evaluate whether there exists an optimal exercise amount related to the best cardiovascular outcomes, we analyzed the hazards of cardiovascular morbidities according to stratified energy expenditure. This analysis was performed in the new exerciser (No to Yes) group to measure the influence of exercise introduction by isolating other healthy confounders that might contribute to the maintenance of exercise. Energy expenditure denotes the "minimum" amount of energy consumption calculated from a self-reported survey question that asks the frequency of each intensity of exercise as a unit of least duration. To calculate energy expenditure, we rated light, moderate, and vigorous intensity exercise as 2.9, 4.0, and 7.0 metabolic equivalents of task (METs), respectively [24]. The total energy expenditure level—the summation of multiplying 2.9, 4.0, and 7.0 METs by frequency of light, moderate, and vigorous exercise together with minimum duration—was stratified into <500, 500–999, 1,000–1,499, and ≥1,500 MET-min/wk in an explorative way.

## Covariates, follow-up, and clinical outcomes

Demographics, comorbidities (hypertension, diabetes mellitus, dyslipidemia, MI, peripheral artery disease [PAD], chronic obstructive pulmonary disease [COPD], cancer, and chronic kidney disease [CKD]), $CHA_2DS_2$-VASc score, medications, laboratory results, anthropometric measurements, responses to the self-reported questionnaire conducted at the health examination after AF diagnosis (index date), and income-based insurance contributions were retrieved to analyze the baseline characteristics of individuals. To evaluate the relationship between exercise habit change and cardiovascular outcomes, we determined incident ischemic stroke and HF using International Classification of Diseases–10th Revision, Clinical Modification (ICD-10-CM) codes in inpatient and outpatient records. The date of death, and ICD-10-CM code recorded for the death, was obtained from NHID and used to investigate all-cause death. We defined a major cardiovascular death as a death from the most frequent 4 major specific diseases that belong to diseases of the circulatory system (I codes in the ICD-10 system), and that most often are the cause of death among Korean AF patients: cerebral infarction (I63), acute MI (I21), sequelae of cerebrovascular disease (I69), and HF (I50) [14].

The follow-up period was defined as the time from the index date to the first occurrence of ischemic stroke, HF, or death or December 31, 2017, whichever came first. Detailed definitions of comorbidities and study outcomes are provided in S1 Table [2,25,26].

## Statistics

Continuous variables are reported as mean ± standard deviation, and categorical variables are reported as number (percentage). We performed inverse probability of treatment weighting (IPTW) using propensity scores derived from all the baseline covariates including demographics, comorbidities, anthropometric measures, lifestyle behaviors, medications, and income status to balance the weights across the 4 study groups. Maximum absolute standardized difference (ASD) < 0.1 (10%) was evaluated as a negligible difference in the baseline covariates between the study groups [27,28]. After IPTW, the weighted incidence rates of ischemic stroke, HF, and all-cause death were computed by dividing each weighted event number by the total follow-up duration and presented as per 1,000 person-years (PY). The risks of the outcomes were computed by weighted Cox proportional hazards models with IPTW.

To compute the most optimal exercise amount for the best cardiovascular outcomes, we defined the persistent non-exerciser category (No to No) as the reference group and computed

a multivariable-adjusted Cox proportional hazards model with 95% confidence intervals (CIs); model 1 was adjusted for age and sex, and model 2 was adjusted for body mass index (BMI), hypertension, diabetes mellitus, dyslipidemia, previous MI, PAD, COPD, cancer, CKD, $CHA_2DS_2$-VASc score, use of oral anticoagulation, use of antiplatelet agent, use of statin, smoking, heavy drinking, and income level in addition to the variables in model 1. All analyses were 2-tailed, and $p$-values $< 0.05$ were considered statistically significant. Data collection and statistical analysis were performed using SAS version 9.4 (SAS Institute, Cary, NC) from January 2020 to September 2020.

### Subgroup and sensitivity analyses

We performed subgroup analyses of the primary outcomes according to sex, age ($<65$, 65 to $<75$, and $\geq75$ years), and $CHA_2DS_2$-VASc score ($<3$ and $\geq3$). The $p$-value for the interaction was estimated to assess the consistency of patterns in the main results among subgroups.

To verify our results' robustness, we performed sensitivity analyses by multivariable Cox proportional hazards models. Model 1 was adjusted for age and sex, and model 2 was adjusted for body measures, comorbidities, $CHA_2DS_2$-VASc score, medications, lifestyle behaviors, and income level in addition to the variables in model 1. Two analyses were performed in response to suggestions from peer review. We further evaluated the risk of the primary outcomes considering comorbidities or frailty using the Charlson Comorbidity Index (CCI), in addition to computing model 2 to adjust for confounding factors that might lead to a change in exercise status. The optimal amount of exercise was also analyzed in the exercise maintainer (Yes to Yes) group, to extend the finding of the most beneficial exercise level in the new exerciser (No to Yes) group.

## Results

### Baseline characteristics

Of the total 66,692 patients with AF, the mean age was 59.5 ± 12.4 years, and 42,410 (63.6%) were men. Comparison between the final study population and the excluded patients, most of whom were excluded due to unavailable health examination data within 2 years both before and after AF diagnosis, is summarized in S2 Table. Patients who received 0 or only 1 health examination were older, were more likely to be female, had more prevalent comorbidities, and had higher $CHA_2DS_2$-VASc scores than those with 2 health examinations.

The final study population consisted of 4 categories: persistent non-exercisers ($n = 20,354$, 30.5%), new exercisers ($n = 11,874$, 17.8%), exercise dropouts ($n = 11,630$, 17.4%), and exercise maintainers ($n = 22,834$, 34.2%). The baseline characteristics of each group are shown in Table 1. New exerciser and exercise maintainer groups tended to be younger, male-predominant, and less affected with comorbidities (especially PAD, COPD, and CKD), and to have lower $CHA_2DS_2$-VASc scores and healthier lifestyle behaviors. We evaluated the differences and balances among multiple groups using the maximum ASDs of all baseline covariates, using a threshold of 0.1 to indicate imbalance. All of the covariates were well balanced after IPTW (ASDs $\leq 0.1$). The multiple comparisons of covariates and the analysis of degrees of balance achievement between the 4 study groups were performed by calculating ASD (presented in S3 Table).

The mean period from the health examination before AF diagnosis (first examination) to the date of AF diagnosis was 1.1 ± 0.5 years, and from AF diagnosis to the health examination after AF diagnosis (second examination) was 0.9 ± 0.5 years. Overall, the time between the 2 consecutive biennial health examinations was 2.0 ± 0.5 years.

**Table 1. Baseline characteristics of the total study population according to the change of exercise status.**

| | Total | Before IPTW | | | | | Post IPTW | | | | |
|---|---|---|---|---|---|---|---|---|---|---|---|
| | | Persistent non-exerciser | New exerciser | Exercise drop-outs | Exercise maintainer | Maximum ASD | Persistent non-exerciser | New exerciser | Exercise drop-outs | Exercise maintainer | Maximum ASD |
| **No. of participants (%)** | 66692 | 20354(30.52) | 11874(17.80) | 11630(17.44) | 22834(34.24) | | 20446.1(30.66) | 11870.9(17.80) | 11635.8(17.45) | 22739.0(34.10) | |
| **Age** | 59.50±12.40 | 62.32±12.26 | 59.28±12.37 | 60.49±12.19 | 56.67±11.94 | 0.41 | 59.22±12.91 | 59.53±12.42 | 59.47±12.55 | 59.24±11.90 | 0.02 |
| < 65 years | 41216(61.80) | 10605(52.10) | 7448(62.73) | 6787(58.36) | 16376(71.72) | | 12760.9(62.41) | 7342.0(61.85) | 7147.2(61.42) | 14195.3(62.43) | |
| 65 to < 75 years | 18565(27.84) | 6508(31.97) | 3270(27.54) | 3539(30.43) | 5248(22.98) | | 5446.1(26.64) | 3295.4(27.76) | 3299.8(28.36) | 6549.5(28.80) | |
| ≥ 75 years | 6911(10.36) | 3241(15.92) | 1156(9.74) | 1304(11.21) | 1210(5.30) | | 2239.0(10.95) | 1233.5(10.39) | 1188.8(10.22) | 1994.2(8.77) | |
| **Sex** | | | | | | 0.50 | | | | | 0.00 |
| Male | 42410(63.59) | 10676(52.45) | 7299(61.47) | 7167(61.63) | 17268(75.62) | | 13052.1(63.84) | 7554.5(63.64) | 7401.6(63.61) | 14496.9(63.75) | |
| Female | 24282(36.41) | 9678(47.55) | 4575(38.53) | 4463(38.37) | 5566(24.38) | | 7393.9(36.16) | 4316.4(36.36) | 4234.2(36.39) | 8242.1(36.25) | |
| **Comorbidities** | | | | | | | | | | | |
| Hypertension | 41466(62.18) | 13008(63.91) | 7309(61.55) | 7286(62.65) | 13863(60.71) | 0.07 | 12612.9(61.69) | 7380.8(62.18) | 7236.1(62.19) | 14029.6(61.70) | 0.01 |
| Diabetes mellitus | 12887(19.32) | 4252(20.89) | 2264(19.07) | 2349(20.20) | 4022(17.61) | 0.08 | 3939.7(19.27) | 2295.3(19.34) | 2241.8(19.27) | 4365.0(19.20) | 0.00 |
| Dyslipidemia | 7537(11.30) | 2361(11.60) | 1326(11.17) | 1352(11.63) | 2498(10.94) | 0.02 | 2320.4(11.35) | 1341.5(11.30) | 1317.2(11.32) | 2567.6(11.29) | 0.00 |
| Previous MI | 2221(3.33) | 745(3.66) | 376(3.17) | 387(3.33) | 713(3.12) | 0.03 | 672.9(3.29) | 395.6(3.33) | 385.4(3.31) | 745.5(3.28) | 0.00 |
| PAD | 12053(18.07) | 4164(20.46) | 2171(18.28) | 2289(19.68) | 3429(15.02) | 0.14 | 3655.4(17.88) | 2145.1(18.07) | 2095.3(18.01) | 4053.4(17.83) | 0.01 |
| COPD | 10640(15.95) | 3836(18.85) | 1942(16.36) | 1919(16.50) | 2943(12.89) | 0.16 | 3272.1(16.00) | 1895.7(15.97) | 1852.3(15.92) | 3633.8(15.98) | 0.00 |
| Cancer | 4197(6.29) | 1260(6.19) | 835(7.03) | 752(6.47) | 1350(5.91) | 0.05 | 1294.7(6.33) | 746.1(6.29) | 725.0(6.23) | 1439.1(6.33) | 0.00 |
| CKD (GFR ≤ 60) | 6749(10.12) | 2535(12.45) | 1113(9.37) | 1207(10.38) | 1894(8.29) | 0.14 | 2035.8(9.96) | 1203.0(10.13) | 1174.5(10.09) | 2256.5(9.92) | 0.01 |
| **CHA₂DS₂-VASc score** | | | | | | 0.53 | | | | | 0.02 |
| < 3 | 47027(70.51) | 12289(60.38) | 8431(71.00) | 7903(67.95) | 18404(80.60) | | 14408.8(70.47) | 8392.7(70.70) | 8213.9(70.59) | 16219.1(71.33) | |
| ≥ 3 | 19665(29.49) | 8065(39.62) | 3443(29.00) | 3727(32.05) | 4430(19.40) | | 6037.2(29.53) | 3478.2(29.30) | 3421.9(29.41) | 6519.9(28.67) | |
| **Medications** | | | | | | | | | | | |
| OAC | 11771(17.65) | 3544(17.41) | 2092(17.62) | 2115(18.19) | 4020(17.61) | 0.02 | 3567.6(17.45) | 2100.4(17.69) | 2045.2(17.58) | 3966.8(17.44) | 0.01 |
| Warfarin | 9016(13.52) | 2605(12.80) | 1631(13.74) | 1590(13.67) | 3190(13.97) | 0.03 | 2744.2(13.42) | 1609.5(13.56) | 1566.8(13.46) | 3036.1(13.35) | 0.01 |
| NOAC | 3374(5.06) | 1148(5.64) | 567(4.78) | 646(5.55) | 1013(4.44) | 0.06 | 1020.9(4.99) | 600.4(5.06) | 588.1(5.05) | 1121.9(4.93) | 0.01 |
| Aspirin | 13263(19.89) | 4212(20.69) | 2248(18.93) | 2339(20.11) | 4464(19.55) | 0.04 | 4010.9(19.62) | 2363.9(19.91) | 2306.2(19.82) | 4457.5(19.60) | 0.01 |
| P2Y₁₂ inhibitor | 2834(4.25) | 955(4.69) | 489(4.12) | 498(4.28) | 892(3.91) | 0.04 | 865.6(4.23) | 503.1(4.24) | 491.2(4.22) | 942.9(4.15) | 0.00 |
| Statin | 9865(14.79) | 3240(15.92) | 1730(14.57) | 1712(14.72) | 3183(13.94) | 0.06 | 3013.6(14.74) | 1759.4(14.82) | 1718.9(14.77) | 3359.8(14.78) | 0.00 |
| **Laboratory findings (2nd exam)** | | | | | | | | | | | |
| BMI (kg/m²) | 24.48±3.25 | 24.35±3.43 | 24.49±3.25 | 24.48±3.23 | 24.60±3.08 | 0.08 | 24.47±3.43 | 24.48±3.23 | 24.48±3.23 | 24.45±3.08 | 0.01 |
| Waist circumference (cm) | 84.10±8.98 | 83.81±9.29 | 83.93±9.04 | 84.23±9.01 | 84.38±8.63 | 0.06 | 84.35±9.30 | 84.08±8.98 | 84.26±9.02 | 83.67±8.73 | 0.08 |
| Systolic BP (mmHg) | 125.09±15.01 | 125.78±15.57 | 124.99±15.12 | 125.13±15.06 | 124.49±14.36 | 0.09 | 124.92±15.22 | 125.18±15.10 | 124.93±14.98 | 125.01±14.66 | 0.02 |
| Diastolic BP (mmHg) | 77.19±10.09 | 77.14±10.24 | 77.05±10.05 | 77.11±10.10 | 77.34±9.96 | 0.03 | 77.10±10.19 | 77.15±10.05 | 77.12±10.08 | 77.17±9.98 | 0.01 |
| Fasting glucose (mmol/L) | 5.75±1.42 | 5.77±1.50 | 5.73±1.41 | 5.77±1.43 | 5.73±1.36 | 0.03 | 5.75±1.50 | 5.74±1.42 | 5.75±1.42 | 5.74±1.37 | 0.01 |
| Total cholesterol (mg/dL) | 186.13±39.94 | 186.09±39.72 | 186.31±43.06 | 186.07±41.35 | 186.10±37.65 | 0.01 | 186.13±39.47 | 186.16±43.26 | 186.04±41.10 | 186.21±38.01 | 0.00 |
| eGFR (ml/min/1.73m²) | 83.47±28.31 | 82.78±27.71 | 84.23±30.59 | 83.42±29.49 | 83.72±26.95 | 0.05 | 84.57±27.29 | 83.81±30.22 | 83.88±30.07 | 82.73±26.76 | 0.07 |
| **Alcohol consumption** | | | | | | | | | | | |
| Heavy drinker | 4665(6.99) | 1293(6.35) | 770(6.48) | 781(6.72) | 1821(7.97) | 0.06 | 1410.2(6.90) | 831.9(7.00) | 812.7(6.98) | 1575.5(6.93) | 0.00 |
| **Smoking** | | | | | | 0.38 | | | | | 0.00 |
| Never smoker | 38014(57.00) | 13353(65.60) | 6855(57.73) | 7052(60.64) | 10754(47.10) | | 11628.3(56.87) | 6764.7(56.99) | 6628.3(56.97) | 12941.2(56.91) | |
| Ex-smoker | 18229(27.33) | 3936(19.34) | 3125(26.32) | 2806(24.13) | 8362(36.62) | | 4757.3(23.27) | 3166.6(26.68) | 3015.0(25.91) | 6931.9(30.48) | |
| Current smoker | 10449(15.67) | 3065(15.06) | 1894(15.95) | 1772(15.24) | 3718(16.28) | | 4060.5(19.86) | 1939.6(16.34) | 1992.4(17.12) | 2865.9(12.60) | |
| Low income* | 10817(16.22) | 3736(18.36) | 1943(16.36) | 1950(16.77) | 3188(13.96) | 0.12 | 3311.1(16.19) | 1926.2(16.23) | 1882.3(16.18) | 3658.0(16.09) | 0.00 |

Data are presented as means ± standard deviation or No. (Percentages).

Percentages may not total 100 because of rounding.

Abbreviation: IPTW, inverse probability of treatment weighting; ASD, absolute standardized difference; MI, myocardial infarction; PAD, peripheral artery disease; COPD, chronic obstructive pulmonary disease; CKD, chronic kidney disease; OAC, oral anticoagulant; BMI, body mass index; BP, blood pressure; LDL, low-density lipoprotein; HDL, high-density lipoprotein; eGFR, estimated glomerular filtration rate.

* Low income denotes income in the lowest 20% among the entire Korean population; individuals with low income are supported by the medical aid program.

## Change in exercise status and ischemic stroke, HF, and all-cause death

Mean follow-up durations for ischemic stroke, HF, and all-cause death were 3.4 ± 2.0, 3.0 ± 2.0, and 3.4 ± 1.9 years, respectively. Table 2 provides the weighted event numbers, weighted incidence rates (per 1,000 PY), and weighted hazard ratios (HRs) of ischemic stroke, HF, and all-cause death after IPTW.

Performing exercise any time before or after AF diagnosis was related to a lower risk of mortality compared to those without exercise: the HR (95% CI) was 0.82 (0.73–0.91) for new exercisers, 0.83 (0.74–0.93) for exercise dropouts and 0.61 (0.55–0.67) for exercise maintainers ($p < 0.001$).

Initiating and maintaining exercise after AF diagnosis was associated with a lower risk of HF compared to the persistent non-exerciser group: the HR (95% CI) was 0.95 (0.90–0.99) for the new exerciser group and 0.92 (0.88–0.96) for the exercise maintainer group ($p < 0.001$). However, the exercise dropouts exhibited a similar risk of HF as the persistent non-exerciser group.

For ischemic stroke, participating in exercise was associated with 10%–14% lower risk compared to the persistent non-exerciser group, but the precision in the estimates of HRs was statistically insignificant: the HR (95% CI) was 0.90 (0.79–1.03) for the new exerciser group and 0.86 (0.77–0.96) for the exercise maintainer group ($p = 0.057$). The weighted incidence rates and risks of cardiovascular outcomes according to the change of exercise status are illustrated in Fig 2.

Since the effect size of lowered estimates of HRs in all-cause death was larger than that of ischemic stroke and HF, we investigated whether the lower risk of all-cause death was

**Table 2. Hazard ratios with 95% confidence intervals for ischemic stroke, heart failure, and all-cause death according to change of exercise status.**

| Outcome and exercise group | Number of individuals | Number of events* | IR* (per 1,000 PY) | HR* (95% CI) or p-value |
|---|---|---|---|---|
| **Ischemic stroke** | | | | |
| Persistent non-exercisers | 20,354 | 633.22 | 9.13 | 1 (Ref.) |
| New exercisers | 11,874 | 331.35 | 8.25 | 0.90 (0.79–1.03) |
| Exercise dropouts | 11,630 | 323.73 | 8.15 | 0.89 (0.78–1.02) |
| Exercise maintainers | 22,834 | 597.22 | 7.87 | 0.86 (0.77–0.96) |
| | | | | $p = 0.057$ |
| **Heart failure** | | | | |
| Persistent non-exercisers | 20,354 | 4,132.32 | 66.51 | 1 (Ref.) |
| New exercisers | 11,874 | 2,279.61 | 62.95 | 0.95 (0.90–0.99) |
| Exercise dropouts | 11,630 | 2,368.37 | 66.53 | 1 (0.95–1.05) |
| Exercise maintainers | 22,834 | 4,201.33 | 61.32 | 0.92 (0.88–0.96) |
| | | | | $p < 0.001$ |
| **All-cause death** | | | | |
| Persistent non-exercisers | 20,354 | 981.20 | 13.87 | 1 (Ref.) |
| New exercisers | 11,874 | 462.96 | 11.32 | 0.82 (0.73–0.91) |
| Exercise dropouts | 11,630 | 465.04 | 11.49 | 0.83 (0.74–0.93) |
| Exercise maintainers | 22,834 | 651.07 | 8.42 | 0.61 (0.55–0.67) |
| | | | | $p < 0.001$ |

*p*-Values were evaluated by the likelihood ratio test.

HR, hazard ratio; IR, incidence rate; PY, person-years.

*Weighted event numbers and weighted IRs were computed after inverse probability of treatment weighting (IPTW). The HRs were computed by weighted Cox proportional hazards models with IPTW.

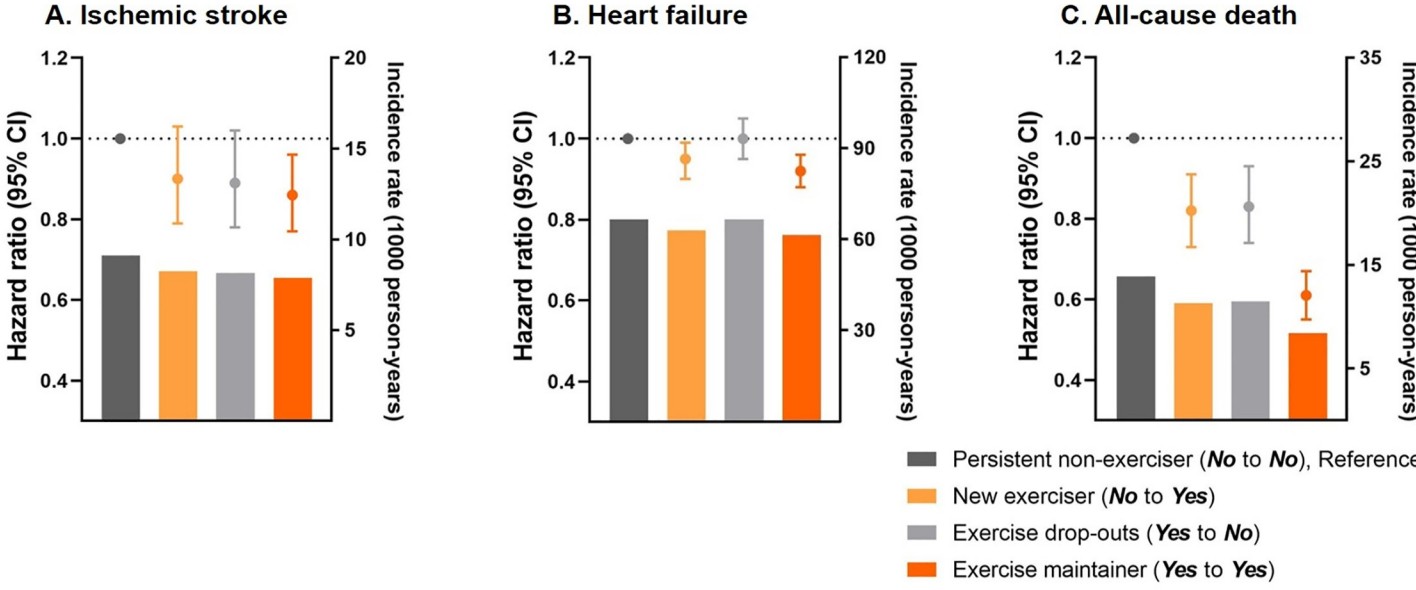

**Fig 2. Hazard ratios with 95% confidence intervals for ischemic stroke, heart failure, and all-cause death according to the change of exercise status.** The bars denote weighted incidence rates, the dots denote hazard ratios, and the whiskers denote 95% confidence intervals computed by weighted Cox proportional hazards models with inverse probability of treatment weighting.

indirectly attributable to the unadjusted confounders such as comorbidities or frailty, rather than possible beneficial effects of exercise. The risk of death from major cardiovascular events (major cardiovascular death) was lower in the exercise maintainers (HR 0.66, 95% CI 0.49–0.90, $p = 0.040$) compared to the persistent non-exercisers. Both the new exercisers (HR 0.74, 95% CI 0.52–1.06) and the exercise dropouts (HR 0.71, 95% CI 0.49–1.02) tended to show a lower HR without statistical significance, which might be due to low event numbers (S4 Table). However, a lower risk of major cardiovascular death in the exercise maintainer group and lower estimates of HRs in the new exerciser and the exercise dropout groups were consistent with the positive risk benefits in all-cause death ($p = 0.040$).

## The relationship between energy expenditure and the risk of cardiovascular outcomes

Multivariable-adjusted Cox proportional hazards analysis of ischemic stroke, HF, and all-cause death in the new exerciser group (No to Yes) according to the stratified amount of exercise is presented in S5 Table. Exercising 1,000–1,499 MET-min/wk (1,208.1 ± 159.7 MET-min/wk) was consistently associated with a lower risk of adverse cardiovascular outcomes (adjusted HR 0.77, 95% CI 0.60–0.99, $p = 0.229$, for ischemic stroke; adjusted HR 0.90, 95% CI 0.82–0.99, $p = 0.166$, for HF) and all-cause death (adjusted HR 0.68, 95% CI 0.54–0.84, $p = 0.002$) compared to the persistent non-exerciser group. Lower risks of ischemic stroke, HF, and all-cause death were correlated with energy expenditure increment, but the associations became attenuated in the group with exercise amount at or above 1,500 MET-min/wk (Fig 3).

## Subgroup and sensitivity analyses

Subgroup analysis was performed by sex, age, and $CHA_2DS_2$-VASc score (S6–S8 Tables). For ischemic stroke, there were no significant statistical interactions for sex, age, and $CHA_2DS_2$-VASc score. The incidence of HF showed an interaction with age, potentially due to an

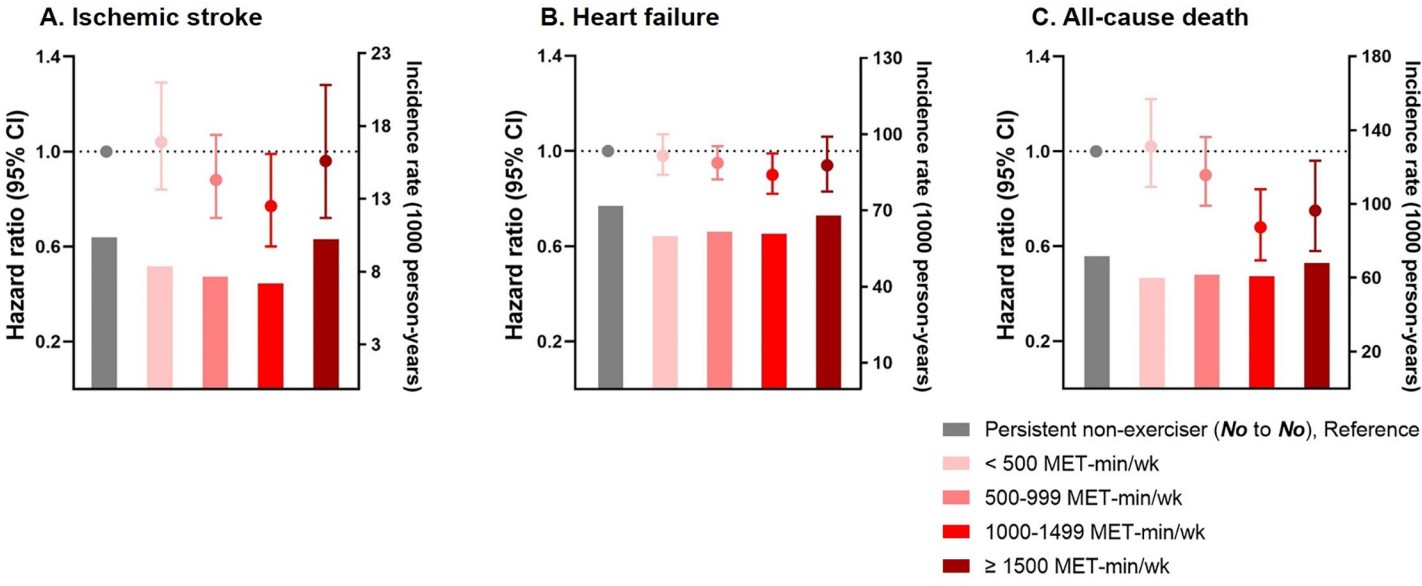

**Fig 3. Hazard ratios with 95% confidence intervals for ischemic stroke, heart failure, and all-cause death in the new exerciser group (No to Yes) according to energy expenditure (MET-min/wk).** The bars denote weighted incidence rates, the dots denote hazard ratios, and the whiskers denote 95% confidence intervals computed by weighted Cox proportional hazards models with inverse probability of treatment weighting. MET, metabolic equivalent of task.

augmented lower risk of HF in the exercise maintainer group among those aged 75 years and older (*p* for interaction = 0.019). The risk of all-cause death presented an interaction with sex that might be explained by greater benefits of exercise in male patients (*p* for interaction = 0.048).

Sensitivity analyses validating the risks of primary outcomes according to the change of exercise habits estimated by multivariable-adjusted Cox proportional hazards models are presented in S9 Table. The risks of ischemic stroke, HF, and all-cause death adjusted by model 2 and CCI were consistent with those in the main analysis. Indeed, S1 Fig shows various adjustment models and corresponding HRs, showing consistent outcomes irrespective of the variables incorporated in the adjustments. The HRs were only different in subtle ways between the models. Another sensitivity analysis verifying whether there exists a dose–response relationship between energy expenditure and the risk of cardiovascular outcomes was performed in the exercise maintainer group (Yes to Yes) and is presented in S10 Table. The levels of energy expenditure showing the lowest estimates of HRs were all different across ischemic stroke, HF, and all-cause death, unlike the consistent level in the new exerciser group (No to Yes).

## Discussion

In this nationwide population-based cohort study, our principal findings are as follows. First, initiation and maintenance of exercise after AF diagnosis were associated with a significantly lower risk for HF and all-cause death. Interestingly, exercise dropout was associated with a lower risk of all-cause death as well. Second, the estimates of HRs of ischemic stroke were lower in AF patients with any lifetime experience of exercise before and after AF diagnosis, though the association was statistically insignificant. Third, performing exercise at an energy expenditure level of 1,000–1,499 MET-min/wk (moderate intensity exercise 170–240 min/wk or vigorous intensity exercise 140–210 min/wk) was consistently associated with lower risks of ischemic stroke, HF, and all-cause death in the new exerciser group. This estimated range of energy expenditure is in accord with the previously recommended volume of exercise in AF

patients [11], yet slightly higher than the minimum leisure-time physical activity suggested for the general population [29].

AF is associated with an increased risk of death and other cardiovascular events [4], which confers a major healthcare burden. A recent cohort study in men demonstrated the inverse association between fitness and mortality [30,31], and the same relationship was reproduced in patients with AF. For example, in the HUNT3 study, physical activity levels in AF patients showed an inverse correlation with risks of all-cause and CVD mortality, as well as the risk of CVD morbidity [32]. Also, the EORP-AF registry reported a lower risk of all-cause death in AF patients engaging in regular exercise at 1-year follow-up and showed an inverse association of cardiovascular outcomes with exercise levels [33]. In addition to there being a mortality benefit of anticoagulation therapy in AF patients [34], our results provide evidence of a mortality benefit of exercise in AF patients as a part of a holistic approach to AF care [15]. Moreover, the benefit of exercise with respect to all-cause death was robust even after evaluating the association by another statistical method and adjusting the model, including as many covariates as possible. Also, the risk benefit for major cardiovascular death, using the 4 most frequent major CVDs explaining deaths of AF patients in Korea [13,14,35–37], presented a pattern consistent with that of all-cause death; thus, the lowered risk of all-cause death might come from the positive effects of exercise.

HF is closely linked to AF by sharing common predisposing conditions, and the coexistence of AF and HF is associated with adverse prognosis [38]. Newly diagnosed AF increases the risk of HF, and individuals with concomitant AF and HF are at a greater risk of all-cause mortality and have higher hospital admission rates [39–41]. Exercise is known to have a protective effect in preventing HF and is beneficial to patients with already established HF as well [42]. Also, moderate and high exercise levels are associated with a lower risk of HF regardless of BMI in the general population [43]. Possible mechanisms contributing to the favorable result regarding long-term HF risk include the reduction of subclinical myocardial damage and abnormal left ventricle remodeling [44,45]. Our study extends the previous findings of exercise in HF patients to patients with AF, whereby those with regular moderate exercise after AF diagnosis showed 5% to 8% lower risk of HF development. Interestingly, the individuals 75 years and older showed even greater benefits of exercise for the risk of HF, suggesting that assessment of daily physical activities and recommendations on exercise are still important for elderly individuals with newly diagnosed AF.

On the other hand, we found that the association between exercise and the risk of stroke was not statistically significant. Increased physical activity lowers stroke risk in the general population [46], but this association was conflicting in patients with AF among previous publications. The HUNT3 study showed an insignificantly lower risk of stroke among patients with AF fulfilling physical activity guidelines, consistent with our study [32]. Meanwhile, another study of cross-country skiers demonstrated that endurance exercise in AF patients is associated with a lower risk of stroke [47]. Moreover, an inverse association between cardiorespiratory fitness and the risk of stroke in patients with AF also has been reported [48]. The relatively low event rate of ischemic stroke compared to HF and all-cause death in our study might attribute to the lack of statistical power; thus, further research needs to address the association between exercise and the risk of ischemic stroke.

The amount of exercise and the risk of AF show a J-shaped relationship; for example, endurance exercise is known to increase the risk of AF [49]. However, no current data provide the specific intensity and amount of exercise that could be recommended to AF patients to benefit cardiovascular outcomes. We not only showed an association between exercise habit change and the clinical outcomes (ischemic stroke, HF, and mortality) in AF patients, we tried to suggest an appropriate amount of exercise. We found that exercising 1,000–1,499 MET-min/wk was associated with lower event rates for stroke, HF, and mortality in the new exerciser group. This level of energy expenditure corresponds to performing moderate intensity

exercise 170–240 min/wk and is comparable to the amount recommended to adults aged 18–64 years with or without chronic conditions: 150–300 min/wk of moderate intensity aerobic physical activity, for substantial health benefits according to WHO guidelines on physical activity and sedentary behavior [50]. Interestingly, the association appeared to be attenuated in the range of exercise ≥1,500 MET-min/wk rather than persistently showing a direct inverse correlation. The mechanisms for intensive exercise to increase the risk of AF are diverse, including changes in autonomic activation [51,52], atrial dilatation [53,54], fibrosis, and inflammation [55,56]. Similar physiological changes would explain the attenuated risk benefit in AF patients with exercise level ≥ 1,500 MET-min/wk across cardiovascular outcomes, and particularly in all-cause death. However, considering the wide 95% CIs in patients with ≥1500 MET-min/wk, the small patient numbers in this group might be underpowered. Besides, the optimal amount of exercise in the exercise maintainer group showed different energy levels having the lowest estimates of HRs for each cardiovascular outcome. We assume that accumulated and prolonged exercise would have influenced the clinical outcomes with different magnitudes of impacts. Taking the evidence together, this exercise dose–response relationship should be interpreted cautiously due to several limitations, including the explorative stratification of exercise and the lack of statistical power. Our quantitative analysis could give a clue for future prospective studies investigating the optimal intensity and amount of exercise in arrhythmia patients.

Exercise has been widely documented to modify potential contributors to AF, such as obesity, hypertension, and diabetes, by improving weight, glucose and lipid control, and endothelial function, and lowering resting heart rate and blood pressure [57–59]. Not only can risk factors for AF but also cardiac structure and function be modified by exercise. For example, Malmo et al. [11] found that, in patients with AF, the exercise group was associated with improved left atrial and ventricular function indices. These mechanisms might contribute to the noticeably lower risk of HF and all-cause mortality (and possibly a lower HR for ischemic stroke) in our study in the new exerciser and exercise maintainer groups compared to the persistent non-exerciser group. Also, arrhythmogenic risk factors and cardiac alterations through exercise might exert continued advantageous effects even in the exercise dropout group, resulting in lower mortality. Of note, we categorized the study population by the change of exercise status after diagnosis of AF; thus, the link between exercise and cardiovascular outcomes could be analyzed according to new exerciser or exercise maintainer status, which could differentiate the benefit of exercise per se from an accumulated gain of exercise and other healthy confounders that enable maintenance of exercise.

A recent large prospective study quantifying daily physical activity among individuals using electronic wearable devices found that individuals with AF engage in significantly less average daily physical activity than those without AF [60]. As we have demonstrated, the influence of exercise mitigated the risks of HF, all-cause death, and probably ischemic stroke in patients with AF, and thus, exercise should be an essential component of nonpharmacological management of AF. Indeed, exercise is part of the lifestyle recommendations in the ABC (Atrial fibrillation Better Care) pathway of holistic or integrated AF management and has been shown in the prospective mAFA-II randomized trial to improve overall outcomes [61]. The ABC pathway refers to the following: "A" for avoid stroke/anticoagulation, "B" for better symptom management, and "C" for cardiovascular and comorbidity management, including lifestyle changes [15].

## Limitations

Several limitations are evident in our study. First, the self-reported nature of the health examination survey asking about exercise frequency would be influenced by recall bias. Second, the

minimum energy expenditure estimated from the 3 questions is limited in its ability to reflect the actual amount of exact exercise level; it would not be possible to precisely quantify exercise (e.g., using pedometers) in this population-based study. Due to limited data about exact exercise volume from the questionnaire, the relationship between endurance exercise and cardiovascular outcomes could not be evaluated. Third, unavailable confounding factors might not be fully adjusted for. Fourth, the type and burden of AF were not available; thus, the contribution of these factors to the outcomes could not be analyzed. Fifth, potential change in exercise status from the index date to the end of follow-up might introduce bias in the associations. Sixth, the study was conducted in an Asian population; therefore, physiological and cultural differences in exercise behavior or biological response to exercise might limit the external generalizability to other ethnic groups. Seventh, the inclusion of only patients who had 2 consecutive health examinations before and after AF diagnosis would cause some selection bias. Our results would have limited evidence in those who could not receive health screening due to underlying diseases; thus, the results should be interpreted and applied cautiously to the general AF population. Lastly, this was not a randomized controlled study, and causality and the direct mechanistic link between exercise and cardiovascular outcomes cannot be determined. However, there may be an ethical issue with study designs restricting exercise (or stopping any exercise), which would make it difficult to include as part of a randomized controlled study. Instead, our data from this large population-based research provide potential evidence for the beneficial effects of exercise on clinical outcomes as part of holistic AF management.

## Conclusion

In this nationwide population-based cohort study, initiating or continuing regular exercise after AF diagnosis was associated with lower risks of HF and mortality. Regular moderate exercise 170–240 min/wk might be an optimal amount for deriving maximal cardiovascular benefits in patients who initiate exercise after AF diagnosis, but further prospective research needs to delineate the specific optimal energy expenditure. Timely encouragement of exercise to overcome misguided fears of arrhythmogenic and sequential adverse outcomes in AF patients would be warranted.

## Supporting information

**S1 STROBE Checklist.**
(DOC)

**S1 Fig. Hazard ratios with 95% confidence intervals for ischemic stroke, heart failure, and all-cause death according to the change of exercise status calculated from the various multivariable-adjusted Cox proportional hazards models.** CI, confidence interval. Model 1 adjusted for age and sex. Model 2 adjusted for body mass index (BMI), smoking, heavy drinking, and low income in addition to the variables in model 1. Model 3 adjusted for diabetes mellitus, hypertension, dyslipidemia, and previous myocardial infarction (MI) in addition to the variables in model 2. Model 4 adjusted for peripheral artery disease (PAD), chronic obstructive pulmonary disease (COPD), cancer, and chronic kidney disease (CKD) in addition to the variables in model 3. Model 5 adjusted for $CHA_2DS_2$-VASc score in addition to the variables in model 4. Model 6 adjusted for use of oral anticoagulation (OAC), use of antiplatelet agent, and use of statin in addition to the variables in model 5. $p$-Values were evaluated by the likelihood ratio test. The dots denote hazard ratios, and the whiskers denote 95% confidence intervals computed by multivariable Cox proportional hazards models.
(DOCX)

**S1 Table. Definition of covariates and outcomes.** CKD, chronic kidney disease; COPD, chronic obstructive pulmonary disease; MI, myocardial infarction; N/A, not applicable; PAD, peripheral artery disease. *Combination: 1 = ICD-10-CM code and medication; 2 = number of diagnosis; and 3 = diagnostic tests or treatment.
(DOCX)

**S2 Table. Baseline characteristics of newly diagnosed AF patients according to the number of health examinations before and after their AF diagnosis.** Data are presented as mean ± standard deviation or number (percentage). Percentages may not total 100 because of rounding. AF, atrial fibrillation; COPD, chronic obstructive pulmonary disease; DOAC, direct oral anticoagulant; HF, heart failure; MI, myocardial infarction; PAD, peripheral artery disease. *Low income denotes income in the lowest 20% among the entire Korean population; individuals with low income are supported by the medical aid program.
(DOCX)

**S3 Table. The multiple comparisons between the study groups presented as absolute standardized differences.** ASD, absolute standardized difference; BMI, body mass index; BP, blood pressure; CKD, chronic kidney disease; COPD, chronic obstructive pulmonary disease; eGFR, estimated glomerular filtration rate; HDL, high-density lipoprotein; LDL, low-density lipoprotein; MI, myocardial infarction; OAC, oral anticoagulant; PAD, peripheral artery disease. *Group A denotes persistent non-exercisers, group B denotes new exercisers, group C denotes exercise dropouts, and group D denotes exercise maintainers. #Low income denotes income in the lowest 20% among the entire Korean population; individuals with low income are supported by the medical aid program.
(DOCX)

**S4 Table. Hazard ratios with 95% confidence intervals for major cardiovascular death according to the change of exercise status.** CI, confidence interval; HR, hazard ratio; IR, incidence rate; PY, person-years. *Major cardiovascular deaths are defined as deaths due to cerebral infarction (I63), acute myocardial infarction (I21), sequelae of cerebrovascular disease (I69), and heart failure (I50). Weighted event numbers and weighted IRs were computed after inverse probability of treatment weighting. The HRs were computed by weighted Cox proportional hazards models with inverse probability of treatment weighting. *p*-Values were evaluated by the likelihood ratio test.
(DOCX)

**S5 Table. Hazard ratios with 95% confidence intervals for ischemic stroke, heart failure, and all-cause death in the new exerciser group (No to Yes) according to the energy expenditure (MET-min/wk).** Exercise dose presented as mean ± SD. CI, confidence interval; HR, hazard ratio; IR, incidence rate; MET, metabolic equivalent of task; PY, person-years. Model 1 adjusted for age and sex. Model 2 adjusted for age, sex, body mass index (BMI), hypertension, diabetes mellitus, dyslipidemia, previous myocardial infarction (MI), peripheral artery disease (PAD), chronic obstructive pulmonary disease (COPD), cancer, chronic kidney disease (CKD), $CHA_2DS_2$-VASc score, use of oral anticoagulation (OAC), use of antiplatelet agent, use of statin, smoking, heavy drinking, and low income. *p*-Values were evaluated by the likelihood ratio test.
(DOCX)

**S6 Table. Hazard ratios with 95% confidence intervals for ischemic stroke, heart failure, and all-cause death according to the change of exercise status and sex.** CI, confidence interval; HR, hazard ratio; IR, incidence rate; PY, person-years. Weighted event numbers and weighted IRs were computed after inverse probability of treatment weighting. The HRs were

computed by weighted Cox proportional hazards models with inverse probability of treatment weighting. *p*-Values were evaluated by the likelihood ratio test.
(DOCX)

**S7 Table. Hazard ratios with 95% confidence intervals for ischemic stroke, heart failure, and all-cause death according to the change of exercise status and age.** CI, confidence interval; HR, hazard ratio; IR, incidence rate; PY, person-years. Weighted event numbers and weighted IRs were computed after inverse probability of treatment weighting. The HRs were computed by weighted Cox proportional hazards models with inverse probability of treatment weighting. *p*-Values were evaluated by the likelihood ratio test.
(DOCX)

**S8 Table. Hazard ratios with 95% confidence intervals for ischemic stroke, heart failure, and all-cause death according to the change of exercise status and CHA2DS2-VASc score.** CI, confidence interval; HR, hazard ratio; IR, incidence rate; PY, person-years. Weighted event numbers and weighted IRs were computed after inverse probability of treatment weighting. The HRs were computed by weighted Cox proportional hazards models with inverse probability of treatment weighting. *p*-Values were evaluated by the likelihood ratio test.
(DOCX)

**S9 Table. Hazard ratios with 95% confidence intervals for ischemic stroke, heart failure, and all-cause death according to the change of exercise status calculated from the multivariable-adjusted Cox proportional hazards model.** CI, confidence interval; HR, hazard ratio; IR, incidence rate; PY, person-years. Model 1 adjusted for age and sex. Model 2 adjusted for age, sex, body mass index (BMI), hypertension, diabetes mellitus, dyslipidemia, previous myocardial infarction (MI), peripheral artery disease (PAD), chronic obstructive pulmonary disease (COPD), cancer, chronic kidney disease (CKD), $CHA_2DS_2$-VASc score, use of oral anticoagulation (OAC), use of antiplatelet agent, use of statin, smoking, heavy drinking, and low income. Model 3 adjusted for age, sex, BMI, hypertension, diabetes mellitus, dyslipidemia, previous MI, PAD, COPD, cancer, CKD, $CHA_2DS_2$-VASc score, use of OAC, use of antiplatelet agent, use of statin, smoking, heavy drinking, low income, and Charlson Comorbidity Index (CCI). *p*-Values were evaluated by the likelihood ratio test.
(DOCX)

**S10 Table. Hazard ratios with 95% confidence intervals for ischemic stroke, heart failure, and all-cause death for exercise maintainer group (Yes to Yes) according to the energy expenditure (MET-min/wk).** CI, confidence interval; HR, hazard ratio; IR, incidence rate; MET, metabolic equivalent of task; PY, person-years. Model 1 adjusted for age and sex. Model 2 adjusted for age, sex, body mass index (BMI), hypertension, diabetes mellitus, dyslipidemia, previous myocardial infarction (MI), peripheral artery disease (PAD), chronic obstructive pulmonary disease (COPD), cancer, chronic kidney disease (CKD), $CHA_2DS_2$-VASc score, use of oral anticoagulation (OAC), use of antiplatelet agent, use of statin, smoking, heavy drinking, and low income. *p*-Values were evaluated by the likelihood ratio test.
(DOCX)

## Author Contributions

**Conceptualization:** Hyo-Jeong Ahn, So-Ryoung Lee, Jae-Hyun Lim, Jun-Pil Yun, Soonil Kwon, Gregory Y. H. Lip.

**Data curation:** Hyo-Jeong Ahn, So-Ryoung Lee, Jin-Hyung Jung.

**Formal analysis:** Hyo-Jeong Ahn, So-Ryoung Lee, Kyung-Do Han, Jin-Hyung Jung, Jae-Hyun Lim, Jun-Pil Yun, Soonil Kwon.

**Investigation:** Hyo-Jeong Ahn.

**Methodology:** Hyo-Jeong Ahn, So-Ryoung Lee, Kyung-Do Han, Jin-Hyung Jung, Jae-Hyun Lim, Jun-Pil Yun, Soonil Kwon.

**Project administration:** Eue-Keun Choi.

**Resources:** Eue-Keun Choi.

**Software:** Kyung-Do Han.

**Supervision:** Eue-Keun Choi, Seil Oh, Gregory Y. H. Lip.

**Validation:** Hyo-Jeong Ahn, So-Ryoung Lee, Kyung-Do Han, Jin-Hyung Jung, Seil Oh.

**Visualization:** Hyo-Jeong Ahn, So-Ryoung Lee.

**Writing – original draft:** Hyo-Jeong Ahn, So-Ryoung Lee.

**Writing – review & editing:** Hyo-Jeong Ahn, So-Ryoung Lee, Eue-Keun Choi, Seil Oh, Gregory Y. H. Lip.

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
