## [Editor Report · Decision Letter 0]

14 Jul 2020

Dear Dr Choi, 

Thank you for submitting your manuscript entitled "Exercise habits and the risk of stroke, heart failure, and mortality in patients with incident atrial fibrillation: a nationwide population-based cohort study" for consideration by PLOS Medicine.

Your manuscript has now been evaluated by the PLOS Medicine editorial staff [as well as by an academic editor with relevant expertise] and I am writing to let you know that we would like to send your submission out for external peer review.

Kind regards,

Adya Misra, PhD,

Senior Editor

PLOS Medicine

---

## [Decision Letter · Decision Letter 1]

2 Sep 2020

Dear Dr. Choi,

Thank you very much for submitting your manuscript "Exercise habits and the risk of stroke, heart failure, and mortality in patients with incident atrial fibrillation: a nationwide population-based cohort study" (PMEDICINE-D-20-03103R1) for consideration at PLOS Medicine. 

Your paper was evaluated by a senior editor and discussed among all the editors here. It was also evaluated by five independent reviewers, including a statistical reviewer. The reviews are appended at the bottom of this email and any accompanying reviewer attachments can be seen via the link below:

[LINK]

In light of these reviews, I am afraid that we will not be able to accept the manuscript for publication in the journal in its current form, but we would like to consider a revised version that addresses the reviewers' and editors' comments. Obviously we cannot make any decision about publication until we have seen the revised manuscript and your response, and we plan to seek re-review by one or more of the reviewers. 

We expect to receive your revised manuscript by Sep 23 2020 11:59PM. Please email us (plosmedicine@plos.org) if you have any questions or concerns.

We look forward to receiving your revised manuscript. 

Sincerely,

Emma Veitch, PhD

PLOS Medicine

On behalf of Clare Stone, PhD, Acting Chief Editor, 

PLOS Medicine

plosmedicine.org

*In the last sentence of the Abstract Methods and Findings section, please include a brief note about any key limitation(s) of the study's methodology.

*At this stage, we ask that you include a short, non-technical Author Summary of your research to make findings accessible to a wide audience that includes both scientists and non-scientists. The Author Summary should immediately follow the Abstract in your revised manuscript. This text is subject to editorial change and should be distinct from the scientific abstract. Please see our author guidelines for more information: https://journals.plos.org/plosmedicine/s/revising-your-manuscript#loc-author-summary

*Please clarify if the analytical approach reported here corresponded to one laid out in a prospective protocol or analysis plan? Please state this (either way) early in the Methods section.

*We'd suggest ensuring that the study is reported according to the STROBE guideline, and the completed STROBE checklist should be included as Supporting Information with the revised paper. Please add the following statement, or similar, to the Methods: "This study is reported as per the Strengthening the Reporting of Observational Studies in Epidemiology (STROBE) guideline (S1 Checklist)." The STROBE guideline can be found here: http://www.equator-network.org/reporting-guidelines/strobe/. When completing the checklist, please use section and paragraph numbers, rather than page numbers.

Comments from the reviewers:

Reviewer #1: The study by Ahn and coworkers provides "big data" from nationwide population based cohort study on the impact of exercise on clinical sequelae of newly diagnosed atrial fibrillation.

They retrieved data from more than 66'000 South Korean patients how had their first diagnosis of atrial fibrillation between two consecutive biennially health care examinations from the National Health Insurance data.

The categorized these patients according to their exercise behavior before and after the diagnosis into 4 groups. The primary endpoint was the incidence of stroke, heart failure and overall mortality. Furthermore, they strived to identify the optimal amount of exercise per week in order to decrease the clinical endpoints in patients who newly started to exercise.

The paper is very well written and the analysis is based on an impressive number of patients. The results are relevant since they underline the importance of initiating or maintaining regular exercise in order to decrease the incidence of heart failure and overall mortality. Furthermore, it provides some guidance as to the weekly duration of moderate exercise in order to derive the maximum benefit.

Specific comments: 

It needs to be mentioned in the limitation section that these results apply to an Asian population which shows some physiological as well as cultural differences.

The amount of women who do not exercise at all appears to be much higher than in a Western European population.

One of the biggest drawbacks is certainly that we do not have any information on the type and burden of AF. 

Reviewer #2: This is a statistical review of manuscript PMEDICINE-D-20-03103_R1. The manuscript is well-written, well-structured and I must say that Figure 1 is really good because it is easy to understand and it captures the whole study design. I have one question for the authors, two suggestions and a few minor points.

Question:

1/ "Lower risks of ischemic stroke, HF, and all-cause death were correlated with energy expenditure increment, but with a trend towards increased risks at ≥1500 MET min/week group". In eTable2, I've noticed that the sample size for the group >= 1500 is the smallest by far. Is there any biological plausibility for the U-shape for the HR curve in Figure 3 ? The CIs overlap so much that I am not sure one can conclude and mention in the abstract that 1000-1499 is the "best" amount of exercise. What are your thoughts? I would suggest to perhaps tone down the conclusions with respect to this analysis. If anything, I see a downward trend. 

Suggestions:

1/ the HRs that you report in the abstract are adjusted HRs. I think that this fact could be mentioned, e.g adjusted hazard ratio (HR) 0.94

2/ we performed reexamination of outcomes within subgroups of sex, age, and CHA2DS2-VASc score. I think it would be helpful to provide more details on the model. Was the model the same as Model 2, with an interaction term between exercise category and sex, age or the score? 

Minor points: I've noticed two sentences where I think one word was missing (see [xxx])

1/ The amount of energy expenditure [that] maximizes cardiovascular benefits appears to be

2/ Since we evaluated de novo incidence of ischemic stroke and HF after AF diagnosis, patients [with] recorded prior history of ischemic stroke and HF before the index date (health examination date after AF diagnosis) were excluded

Reviewer #3: The study by Ahn et al sought to determine the influence of exercise after atrial fibrillation (AF) diagnosis on risk of stroke, heart failure and all-cause death. The topic and aim are highly relevant given that specific exercise guidelines are lacking for AF patients and there is a paucity of data on cardiovascular outcomes after exercise in established AF. Particularly novel is the use of data on exercise habits before and after first AF diagnosis. The design and results strengthen the idea that exercise is causally related to outcomes, in a relatively short term, and therefore might have a place in secondary prevention and cardiac rehabilitation of these patients. Other strengths of the study are a large population sample, a high event rate and seemingly valid and relevant endpoints. The paper is also well-written and structured. I have only some minor comments that may improve the manuscript:

* Stroke risk is obviously a major issue in AF and exercise may play a role in risk reduction. Unfortunately, this study, similar to the cited study from the HUNT cohort, was not able to provide convincing evidence that exercise adoption or maintenance would greatly reduce the risk of stroke. Still I think the statement that `exercise was associated lower risk of HF, but not stroke` (abstract, conclusion etc.) is too strong and may even be misleading when read briefly and out of context (as many do). The main effect estimates (HR) for stroke in exercisers actually seem in line with, or even stronger, than those for HF although precision obviously is lower. Whether the lack of precision is due to power (events) or other confounders/effect modifiers could be discussed. Also, a certain amount of exercise in new exercisers were even "significant" (Fig 3/eTable 2). I would consider re-wording these conclusions in a neutral language, including line 219 ("impact..not as strong") given that this may be the take-home-message for many readers. Keeping the left x-axis in Fig. 2 consistent across outcomes would also help on correct interpretation.

* Similarly, at p. 9, line 176, it is stated that the highest exercise volumes had a "trend towards increased risks", although the main effect were even lower than the reference, even significantly for deaths. Although I understand the statement refer to the J-shape (or reverse J-shape?) and a comparison to the second highest category, I suggest change to "attenuated risk reduction" or similar.

* PA was reported ~1 year before and after AF diagnosis and that is a strength of the study, but might also be a limitation. I.e. a potential change in PA status from examination to end of follow up might bias the associations and is not mentioned in limitations. In line with that, I`m surprised about the number of "new exercisers", compared to drop-outs after being diagnosed with this exercise-limiting disease (by the way, I would change the word "drop-out" just for the reasons above, as well as "any lifetime experience" at line 188). Can you just comment? Are there any exercise recommendations/rehabilitation standards after AF diagnosis in Korea? 

* The secondary analyses on energy expenditure is interesting and provides novel information for future guidelines. But why were only new exercisers analysed? Would there be differences in maintainers? I think "adopters" or "new exercisers" could be added to the sentence at lines 189-90 since this conclusion about volumes is only valid for previously inactive patients` exercise a year after diagnosis.

* Some spelling and grammar should be corrected, i.e. line 44, 253. Line 273-5; Hard to understand the sentence, do you mean RCT`s is not possible or ethical in this population? 

* Figure 1 seems to be duplicated

Reviewer #4: Hyo-Jeong et al present data obtained from the National Health Information Database from the Republic of Korea. Specifically, they assess the impact of self-reported exercise habits before an initial diagnosis of atrial fibrillation and the impact of a change in exercise habits after that diagnosis using no exercise pre and post diagnosis (persistent non-exerciser) as the baseline risk group for comparison. Outcomes assessed included stroke, CHF and all-cause mortality at follow-up. Additionally, their data allowed an evaluation of the minimum amount of energy expenditure ranges (MET-min/week) in the new exerciser group across the primary outcomes of stroke, CHF and all-cause mortality. This is an interesting study from a large data-set that does contribute to our rapidly growing knowledge set on lifestyle and atrial fibrillation. The authors appropriately note the limitations of self-reported exercise habits which is a common limitation in many such studies. Overall, the study is well written but I have a few comments that should be addressed.

* Line 65: "piled up" is more of a vernacular phrase and should be deleted or reworded

* Line 65: "direct clinical consequences are less likely offered" is vague and I am unclear as to what you intended to state with this phrase

* Line 66 "… limited… " is vague and unclear. There are many exercise recommendations for patients with atrial fibrillation.

* Lines 176-177:".., but with a trend towards increased risks…" I am not proponent of using and stating there were non-significant trends especially in the context of exercise at the end range such as your >1500 MET-min/week group. This is non-significant and should be left as such. In reviewing your eTable 2, all-cause death in this group in model 1 and model 2 still have HR <1 and better than no/minimal exercise. The curve appears to flatten saying that it trends toward increasing is a bold and potentially incorrect statement.

* Line 187: Again there is mention of "trend"

* Lines 232-233: I would caution and recommend not stating "appeared to be J-shape". The curve certainly flattens and this has been shown in previous studies. However, stating a true J-curve exists is difficult given the generally lower events in the end range group with widening of the error bars.

* Lines 235-237: "Similar mechanisms would explain … in patients with AF." See above discussion/comments. 

* Lines 244-245: Again there is a mention of trend (see above)

Reviewer #5: Thanks for giving me the opportunity to review this interesting manuscript titled 'Exercise habits and the risk of stroke, heart failure, and mortality in patients with incident atrial fibrillation: a nationwide population-based cohort study'.

1.Severe selection bias is a big concern. From a big pool of 523174 patients, they end up to a set of 66692 patients for analysis. It is not known how different are these patients from those patients in the original pool. If the authors can provide some comparison, it will be helpful.

2.The commonest cause of death in patients with AF are non-cardiac death, heart failure, and stroke. In this paper, exercise status is not significantly associated with ischemic stroke, and marginally significant associated with heart failure, but very significantly associated with all-cause death. Please explain whether this is due to unadjusted confounders, which means the changes of exercise status is due to other co-morbidities, or frailty of the patients.

3.The intensity and frequency of exercise is self-reported. Whether the questionnaire is validated and how accuracy is the data collected?

4.All the definition of confounders and outcomes are not clearly defined. It would be good to clarify them. 

5.Figure 1 and figure 1 should be combined, as the information of efigure 1 is also important. 

6.efigure 1, the flow chart, all the excluded patients and reason for exclusion should be listed. I guess 523174-19148 patients did not get consecutive health examination?

[LINK]

---

## [Decision Letter · Decision Letter 2]

5 Feb 2021

Dear Dr. Choi,

Thank you very much for submitting your manuscript "Exercise habits and the risk of stroke, heart failure, and mortality in patients with incident atrial fibrillation: a nationwide population-based cohort study" (PMEDICINE-D-20-03103R2) for consideration at PLOS Medicine. 

[LINK]

In light of these reviews, I am afraid that we will not be able to accept the manuscript for publication in the journal in its current form, but we would like to consider a revised version that addresses the reviewers' and editors' comments. Obviously we cannot make any decision about publication until we have seen the revised manuscript and your response, and we plan to seek re-review by one or more of the reviewers. 

Thank you for addressing the previously stated issues. Before we proceed, please address the following issues:

• Please mention country in title and abstract.

• PLOS defines the “minimal data set” to consist of the data set used to reach the conclusions drawn in the manuscript with related metadata and methods, and any additional data required to replicate the reported study findings in their entirety. Authors do not need to submit their entire data set, or the raw data collected during an investigation. Please submit the following data: 1) The values behind the means, standard deviations and other measures reported; 2) The values used to build graphs; and 3) The points extracted from images for analysis.

• Please bullet and trim the author summary - preferably use the active voice in 1-2 points ("We investigated ..." etc).

• Abstract Methods and Findings:

1. Please include the study design, population (summary of demographic details for participants), and setting.

2. Please quantify the main results (please present both 95% CIs and p values).

3. Please include the important variables that are adjusted for in the analyses.

4. Please include the actual amounts and/or absolute risk(s) of relevant outcomes (including NNT or NNH where appropriate), not correlation coefficients

5. Final sentence of the "Methods and findings" subsection of abstract should begin "Study limitations include ..." or similar.

• In the Abstract conclusion, please mention specific implications for policy or clinical care substantiated by the results. Please revise Conclusion, first line (line 50) and instead write, “Initiation or continuing regular exercise after AF was associated with a lower risk of HF and mortality among patients.”

• In Methods and Results please update the following:

1. Early in methods section of main text, please state that there was no protocol or prespecified analysis plan

2. Please provide the actual numbers of events for the outcomes, not just summary statistics or ORs.

3. Please provide p values in addition to 95% Cis

4. When a p value is given, please specify the statistical test used to determine it.

5. Changes in the analysis made in response to peer review comments should be identified as such in the Methods section of the paper, with rationale.

6. Please ensure p<0.0001 is revised to p<0.001

• The following issues have the potential to introduce bias in your study. Please address the following in your analyses or explaining how you addressed potential bias:

1. Reporting bias is a threat to internal validity. Please discuss the potential for systematic differences in responses between those who experienced and did not experience the outcomes of interest. 

2. In your statistical analyses, please use hierarchical/ multilevel models given that nationwide data is likely clustered at various regional levels. The potential clustering of data (e.g., among patients from the same locality or hospital) would result in spurious effect estimates and standard errors.

3. Please employ more robust methods to address unobserved confounding e.g., use propensity score matching or instrumental variables.

4. Please clearly describe how you selected adjustment variables in model 2; most of the included variables are potentially correlated. Directed acyclic graphs are instrumental in making models as parsimonious as possible. 

5. Please state whether you did post-hoc corrections for multiple comparisons in your analyses

• Under the “Subgroup and additional analysis” section, 2nd paragraph (line 249), the term "trend" is used to refer to a nonsignificant P value. The term trend should be used only when the test for trend has been conducted. Please revise accordingly.

• Please replace "subject" with participant, patient, individual, or person.

• Discussion: Line 294: Please change "statistically insignificant" to "not statistically significant" or similar.

• Discussion: Line 355: Please change "western populations" to "populations of high-income countries" or similar.

• Table 2, eTable 4,5,6,7,8, and 9: Please provide p values in addition to the 95% CIs reported.

• Please indicate in the figure caption the meaning of the bars and whiskers in Figure 2 & 3

• In the Reference section, please delete the following statement from reference 42 (line) and paste it under the appropriate COI section:

“www.icmje.org/coi_disclosure.pdf and declare: no support from any organisation for the submitted work; no financial relationships with any organisations that might have an interest in the submitted work in the previous three years; no other relationships or activities that could appear to have influenced the submitted work.”

We expect to receive your revised manuscript by Feb 26 2021 11:59PM. Please email us (plosmedicine@plos.org) if you have any questions or concerns.

We look forward to receiving your revised manuscript. 

Sincerely,

Dr Raffaella Bosurgi, 

executive editor

Medicine 

PLOS Medicine

plosmedicine.org

Comments from the reviewers:

Reviewer #2: Thank you for addressing my previous comments. I do not have any further comments. Best of luck. 

Reviewer #3: All my comments and concerns are taken into account in this manuscript that I think is very much improved. I think the consistency across y-axis A, B, C in Fig. 2 very much helped the interpretation and would consider to do the same with Fig. 3 although I leave the decision to the authors or editors.

Reviewer #5: My comments are all well addressed. I have no further comments.

[LINK]

---

## [Decision Letter · Decision Letter 3]

22 Apr 2021

Dear Dr. Choi,

Thank you very much for re-submitting your manuscript "Exercise habits and the risk of stroke, heart failure, and mortality in patients with incident atrial fibrillation: a Korean nationwide population-based cohort study" (PMEDICINE-D-20-03103R3) for review by PLOS Medicine.

I have discussed the paper with my colleagues and the academic editor and it was also seen again by two reviewers. I am pleased to say that provided the remaining editorial and production issues are dealt with we are planning to accept the paper for publication in the journal.

[LINK]

We look forward to receiving the revised manuscript by Apr 29 2021 11:59PM.   

Sincerely,

Beryne Odeny, 

Associate Editor 

PLOS Medicine

plosmedicine.org

Requests from Editors:

• Please revise your title according to PLOS Medicine's style. Your title must be nondeclarative. It should begin with main concept if possible. "risk" should be used only if causality can be inferred, i.e., for an RCT. Please place the study design in the subtitle (ie, after a colon). For example, “Exercise habits, stroke, heart failure, and mortality in patients with incident atrial Fibrillation in Korea: a population-based retrospective cohort study”

• Please trim your author summary bullet points

• Please remove NNT values from your results as recommended by reviewer #5. These are not applicable

• The Data Availability Statement (DAS) requires revision. For each data source used in your study: 

• In your methods, please indicate how you accounted for multiple comparisons in your analyses. Did you use the Bonferroni correction or similar?

• Please avoid using causal language such as “risk of”, “impact of”, “effect of” etc, and use “associated with” instead, in keeping with the study design of this work.

Comments from Reviewers:

Reviewer #2: Dear authors, thank you for addressing the comments from previous reviews. I do not have additional comments at this stage. 

Reviewer #5: This R3 manuscript is generally well written. The authors endeavored to address the reviewer's comments and they should be commended for having done a good job. However, I still have 2 major concerns with the manuscript:

1. This is an observational study, which only association can be derived. NNT suggests a causal effect. The authors stated they added the NNTs as the peer reviewer required, but I strongly discourage adding such an interpretation to mislead the readers. 

2. The authors concluded "Regular moderate exercise 170-240 min/week was associated with the maximal benefit on cardiovascular outcomes in patients who initiate exercise after AF diagnosis". I do not agree such an interpretation. from an eyeball impression, the confidence interval of HR for different exercise groups are largely overlapped. Interaction has not ever been tested. It is also not believable 170-240min/week moderate exercise is the best recommendation to AF patients. Moreover, as the data of exercise is self-reported and only roughly classified into different categories, it is not reasonable to evaluate these effect quantitatively.

[LINK]

---

## [Editor Report · Decision Letter 4]

18 May 2021

Dear Dr Choi, 

On behalf of my colleagues and the Academic Editor, Kazem Rahimi, I am pleased to inform you that we have agreed to publish your manuscript "Association between exercise habits and stroke, heart failure, and mortality in Korean patients with incident atrial fibrillation: a nationwide population-based cohort study" (PMEDICINE-D-20-03103R4) in PLOS Medicine.

PRESS

Sincerely, 

Beryne Odeny 

Associate Editor 

PLOS Medicine